# National Program for External Quality Assessment of Chinese Newborn Screening Laboratories

**DOI:** 10.3390/ijns6020038

**Published:** 2020-05-09

**Authors:** Yuxuan Du, Wei Wang, Jiali Liu, Zhixin Zhang, Zhen Zhao, Falin He, Shuai Yuan, Zhiguo Wang

**Affiliations:** 1National Center for Clinical Laboratories, Beijing Hospital, Beijing 100730, China; yxdu@nccl.org.cn (Y.D.); wwang@nccl.org.cn (W.W.); 18311057644@163.com (J.L.); xinshai16@163.com (Z.Z.); 15901515520@139.com (Z.Z.); flhe@nccl.org.cn (F.H.); syuan@nccl.org.cn (S.Y.); 2National Center of Gerontology, Beijing Hospital, Beijing 100730, China; 3Institute of Geriatric Medicine, Chinese Academy of Medical Sciences, Beijing 100730, China

**Keywords:** newborn screening, external quality assessment, coefficient of variation

## Abstract

Objectives: To analyze the coefficient of variation (CV) of external quality assessment (EQA) in Chinese newborn screening (NBS) laboratories. Method: EQA’s robust CV was analyzed by the Clinet-EQA evaluation system. Results: Participating laboratories of the EQA program increased annually. There was more than a 11-fold increase in phenylalanine (Phe) and thyroid stimulating hormone (TSH). It has shown a declining robust CV, which has tended to level off in recent years. The interquartile range (IQR) of Phe and TSH’s robust CV has decreased from 15.5% to 1.5% and from 22.8% to 1.8%, respectively. Compared to bacterial inhibition assay (BIA), the robust CV of Phe has been shown to be relatively reduced in the fluorescence assay and quantitative enzymatic assay (QEA). The robust CV by ELISA was relatively unstable compared to DELFIA and FEIA. In addition, the robust CVs of glucose-6-phosphate dehydrogenase (G6PD) and 17-alpha-hydroxy progesterone (17-OHP) by Genetic Screening Processor (GSP) were lower than other systems. The median of robust CV by non-derivatized MS/MS (Fenghua) in Phe and free carnitine were around 2.2–4.7% and 2.6–5.2%. Conclusion: Neonatal screening has developed rapidly in China and the majority of participant laboratories had satisfactory performance for the quantitative results.

## 1. Introduction

Neonatal or newborn screening (NBS) for a series of disorders such as phenylketonuria (PKU), congenital hypothyroidism (CH), glucose-6-phosphate dehydrogenase (G6PD) deficiency and congenital adrenal hyperplasia (CAH) is widely used in most developed and some developing countries [1]. NBS programs using dried blood spots (DBS) were first developed in the 1960s, inspired by the work of Dr. Robert Guthrie [2]. Screening tests are designed to detect asymptomatic newborns at risk for a disease from those who are not at risk. Effective screening of newborns, combined with follow-up diagnostic confirmatory testing and treatment, helps to prevent morbidity and mortality.

NBS in China started with a pilot study in 1981, and in the following two decades was extended to 31 provinces [3]. All NBS laboratories must participate in external quality assessment (EQA) programs through the Technical Guide to Neonatal Screening (2004), Regulation on Neonatal Screening (2009), and the Technical Guide to Neonatal Screening (Revised edition, 2010) to evaluate the quality of laboratory performance. The EQA challenges are designed to mimic neonatal specimens, and thus should be assayed following the laboratories’ standard operating procedures. The National Center for Clinical Laboratories (NCCL) has gradually launched EQA programs of neonatal screening for analytes since 1998, as the phenylalanine (Phe) and thyroid stimulating hormone (TSH) EQA samples were distributed, following by the first G6PD and 17-alpha-hydroxy progesterone (17-OHP) external quality survey in 2010 [4]. Moreover, the tandem mass spectrometry (MS/MS)-based EQA program was launched on NBS screening to measure amino acid and acylcarnitine by NCCL in 2013 [5,6], as MS/MS become the standard for the screening of disorders of fatty acid oxidation, amino acid metabolism, and organic acidurias [7].

## 2. Materials and Methods

### 2.1. EQA Participanting Laboratories

Clinical laboratories providing neonatal screening services were required to participate in the EQA program by NCCL according to Regulation on Neonatal Screening (2009).

### 2.2. EQA Panel

EQA panels were prepared and each panel contained 5 DBS. One panel of Phe and TSH EQA samples was distributed to participating laboratories in 1999, and subsequently three panels with 15 DBS were distributed annually from 2000 to 2019. Two panels with 10 DBS of G6PD EQA samples were distributed annually from 2012 to 2014, and subsequently three panels were distributed annually from 2015 to 2019. Two panels of 17-OHP EQA samples were distributed annually from 2013 to 2015, and subsequently three panels were distributed annually from 2016 to 2019. Two panels of amino acid and acylcarnitine by MS/MS EQA samples were distributed annually from 2013 to 2019. Testing results were collected by the Clinet-EQA evaluation system V 1.0, designed by NCCL and used in the national EQA program (see http://www.clinet.com.cn/shop/shop).

### 2.3. Statistical Analysis

Data were analyzed using the Clinet-EQA evaluation system V 1.0. The robust coefficient of variation (CV) was calculated by the algorithm A referring to ISO 13528 [8]. The results were expressed as median and interquartile range. The data were analyzed using Student’s *t*-test. A *p*-value less than 0.05 was considered significant.

## 3. Results

### 3.1. Annual Number of Participanting Laboratories

Clinical laboratories providing neonatal screening services participated in the EQA program by NCCL. The neonatal screening EQA by testing Phe and TSH was performed in 1999 for the first time. The G6PD and 17-OHP EQA programs were launched from 2012 and 2013 onwards, respectively. In 2003, NCCL launched the EQA program for the detection of amino acid and acylcarnitine by MS/MS. There was more than a 11-fold increase in the number of participating laboratories of Phe and TSH EQA program over the last 20 years. Meanwhile, there were 155, 191 and 262 laboratories enrolled in G6PD, 17-OHP, amino acid and acylcarnitine in 2019, respectively (Table 1).

### 3.2. Annual Variation of Robust CV

It has been shown that there is a declining robust CV in EQA analytes, which has tended to level off in recent years. For instance, the medians of Phe and TSH have fluctuated around 11.7–17.4% and 8.1–11.0%, respectively, for the last 5 years (Figure 1). Meanwhile, the interquartile range (IQR) of Phe and TSH robust CV has decreased from 15.5% to 1.5% and from 22.8% to 1.8%, respectively (Figure 1a,b). G6PD and 17-OHP, on the other hand, have shown an increasingly robust CV in recent years (Figure 1c,d). Moreover, there was a significant reduction in robust CV of Phe and free carnitine by MS/MS since 2013, which has been shown to reduce the median below 10.0% (Figure 1e,f).

### 3.3. Annual Variation of Grouped Robust CV

To determine the effects of measuring methods, reagents or instruments, data were grouped by measurement system. Compared to bacterial inhibition assay (BIA), the robust CVs of Phe by fluorescence assay and quantitative enzymatic assay (QEA) have been shown to be relatively lower (*p* < 0.001). The robust CVs of fluorescence assay by PerkinElmer (PE), Genetic Screening Processor (GSP), and Fenghua measurement systems were more stable than others (Figure 2a and Figure 3a). Meanwhile, the robust CVs of TSH by DELFIA and FEIA tended to level off for the last 5 years (Figure 3b). The robust CV by ELISA (Unionlock) was relatively unstable (*p* < 0.001), while the median of robust CVs by BioRad ELISA system fluctuated from 6.2 to 10.2% (Figure 2b and Figure 3b). In addition, the robust CVs of G6PD and 17OHP by GSP were much lower than the PE system (*p* < 0.001) and Ani Lab system (*p* < 0.001) (Figure 2c,d and Figure 3c,d). There has been shown to be a high degree of robust CV by derivatized MS/MS (non-kit) in Phe and free carnitine (*p* < 0.001); on the contrary, the median of robust CV by non-derivatized MS/MS (Fenghua) in Phe and free carnitine were around 2.2–4.7% and 2.6–5.2%, respectively (Figure 2e,f and Figure 3e,f).

## 4. Discussion

The improved accuracy could improve the accuracy of diagnoses, which might determine the treatment and thus avoid life-long disability [7]. There are two key strategies in analytic quality, including EQA/proficiency testing (PT) and internal quality control (IQC). The EQA/PT plays an essential role in monitoring and promoting the performance of newborn screening. Covering the majority of NBS laboratories in China, the EQA/PT data were collected via the Clinet-EQA evaluation system by NCCL, and over 80% of the quantitative results are acceptable [9]. In this study, CVs were calculated to monitor the imprecision quality in China. The results of our study indicate that the CVs of EQA analytes varied vastly among different screening laboratories. Comparing the Newborn Screening Quality Assurance Program (NSQAP) by the CDC with the IQR of 17OHP from 19.0% to 37.8% in 2019, for instance, the NBS laboratories in our study revealed a poor performance, indicating the need to enhance the quality and accuracy of newborn screening results. There is a wide gap in comparison with the United States; however, a decrease in the robust CVs was observed for the last 5 years in EQA analytes (Figure 1). Meanwhile the robust CV of G6PD and 17-OHP, on the other hand, went up recently (Figure 1), which might be caused by the unexperienced participating laboratories, or the variation of methods and reagents. This finding was reinforced by analyzing the data in different measurement systems considering the difference in specificity and sensitivity (Figure 2).

Comparative analysis has shown remarkable differences in CVs among methods, reagents, or instruments, suggesting different testing performance to some extent. Fortunately, the measurement systems with poor performance have been improved or out of service afterward, rendering a declining CV in EQA analytes which tend to level off in recent years (Figure 3). To improve the accuracy of detection in neonatal screening, a platform for quantitative EQA failures was developed for clinical laboratories. The laboratories could automatically analyze the trend of the EQA results through the network reporting system; it could also summarize the unsatisfactory EQA performance for certain years and find the potential reasons, including clerical issues, methodological issues, equipment issues, technical issues, PT process issues, and unexplained issues as well.

All EQA/PT challenges were designed to simulate infant blood spots, and thus, the limitation of our study was that the blood spots on the panel were not actually from newborn specimens, which might cause matrix effects [10]. For the EQA/PT challenges, participants also need to report the cut-off value and qualitative results for each analyte tested. It might be complicated to determine whether it is positive or negative, as the quantitative values were around the cut-off values.

In conclusion, neonatal screening has developed rapidly in China and the majority of participating laboratories had satisfactory performance for the quantitative results. However, it still has a certain difference compared to developed countries. Therefore, evaluating and monitoring NBS laboratories’ continuing performance might improve the accuracy of screening results.

## Figures and Tables

**Figure 1 IJNS-06-00038-f001:**
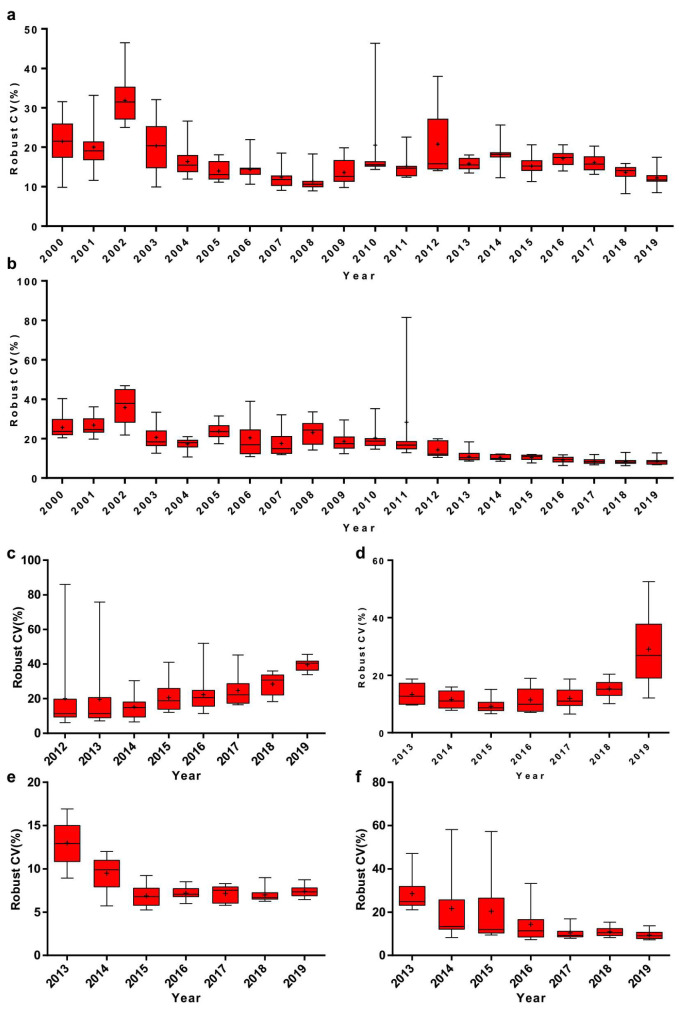
Annual variation of robust coefficient of variation (CV) of (**a**) phenylalanine (Phe), (**b**) thyroid stimulating hormone (TSH), (**c**) glucose-6-phosphate dehydrogenase (G6PD), (**d**) 17-alpha-hydroxy progesterone (17-OHP), (**e**) Phe (MS/MS) and (**f**) free carnitine (MS/MS). Box-and-whiskers plots represent the median and interquartile range (IQR) with whiskers ranging between the min and max; + indicates the mean of robust CV. *n* = 10 or 15 (*n* equals to the number of distributed DBS per year).

**Figure 2 IJNS-06-00038-f002:**
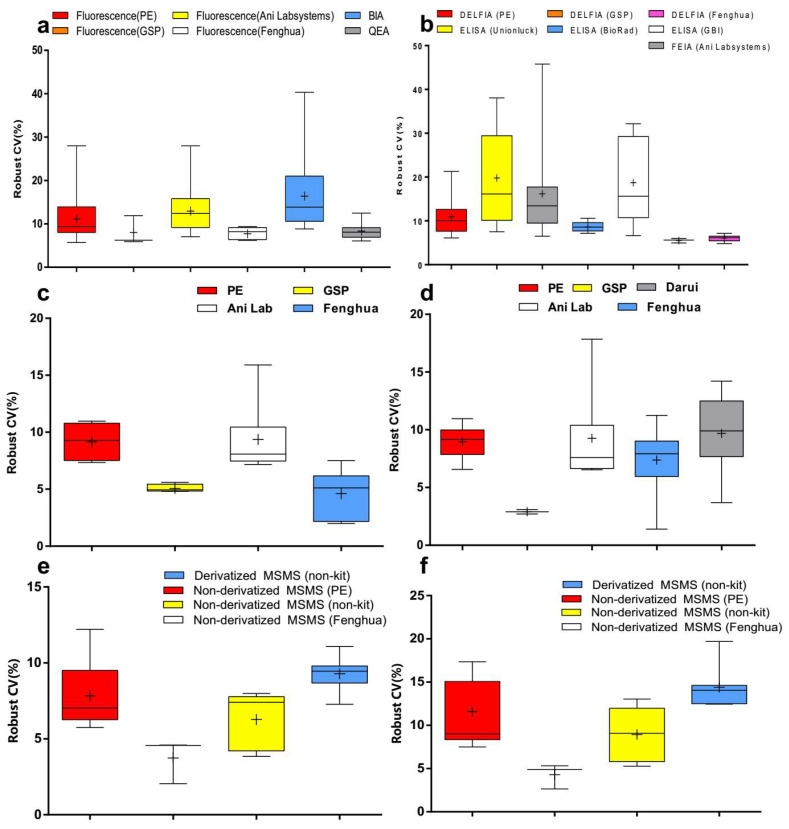
Robust CV grouped by detection system of (**a**) Phe, (**b**) TSH, (**c**) G6PD, (**d**) 17OHP, (**e**) Phe and (**f**) free carnitine. Box-and-whiskers plots represent the median and interquartile range with whiskers ranging between the min and max; + indicates the mean of robust CV. * out-of-range value.

**Figure 3 IJNS-06-00038-f003:**
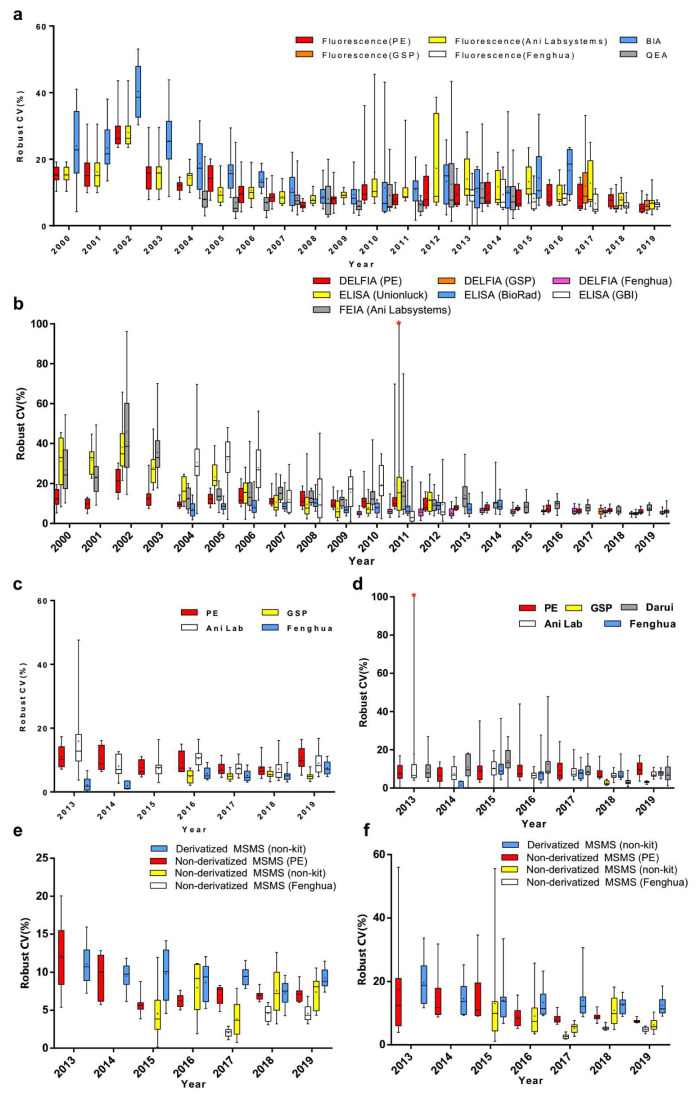
Annual variation of robust CV grouped by detection system of (**a**) Phe, (**b**) TSH, (**c**) G6PD, (**d**) 17OHP, (**e**) Phe and (**f**) free carnitine. Box-and-whiskers plots represent the median and interquartile range with whiskers ranging between the min and max; + indicates the mean of robust CV. * out-of-range value. *n* = 10 or 15 (n equals to the number of distributed DBS per year).

**Table 1 IJNS-06-00038-t001:** Annual number of laboratories participating in external quality assessment (EQA) programs.

Year	Annual Number of Participating Laboratories			
	Phe and TSH	G6PD	17-OHP	Amino Acid and Acylcarnitine (MS/MS)
1999	19			
2000	32			
2001	38			
2002	47			
2003	62			
2004	93			
2005	106			
2006	127			
2007	144			
2008	158			
2009	172			
2010	178			
2011	186			
2012	196	80		
2013	207	75	53	34
2014	212	90	74	45
2015	216	104	98	59
2016	222	113	118	95
2017	238	139	153	156
2018	238	150	170	211
2019	246	155	191	262

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
