# Peer review of "National Program for External Quality Assessment of Chinese Newborn Screening Laboratories"

_2409-515X, 2020, doi:10.3390/ijns6020038_

Round 1

Reviewer 1 Report

Dear authors,

thank you for your description and results of the Chinese EQA Programm. Your results Show very nicely the improvement of the Performance of Chinese NBS laboratories, which also proves that role EQA programmes have.

However, there are a few things, that Need Attention/improvement.

1.) Page 2, line 46: you should write: "... because the Standard for the Screening of disorders of fatty acid oxidation, amino acid metabolism, and organic acidurias ..."

2.) Fig.3 the colours for the different methods/instruments in Fig.3 a-f should be harmonized, and better defined. For example is PE, manuel DELFIA and AutoDelfia ?

3.) The number of laboratories that use the respective method should be indicated

4.) In the results section the author compare and interprete the Performance of different platforms and instruments, however the authors do not provide any calculation or data to prove that the differences they discribe and discuss are really significant

5.) The authors should describe, whether and how the EQA program deals with administrative Errors, i.e. mix-up of results during reporting.

Author Response

Thank you for the constructive criticism and the effort spent by the reviewers improving our manuscript. These comments were very helpful. Included please find a point by point reply to the reviewers’ comments. The manuscript has undergone all requested changes with revisions in the revised manuscript in red.

Point 1: Page 2, line 46: you should write: "... because the Standard for the Screening of disorders of fatty acid oxidation, amino acid metabolism, and organic acidurias ..."

Response 1: Thank you for your kind revision. We have modified it in the manuscript in red.

Point 2: Fig.3 the colours for the different methods/instruments in Fig.3 a-f should be harmonized, and better defined. For example is PE, manuel DELFIA and AutoDelfia ?

Response 2: We appreciate the reviewer’s suggestion. We grouped the data according to the grouping rules in EQA challenges by methods, reagents or instruments, based on the effect of different methods on results, number of participating laboratories etc. Unfortunately, difference between manuel DELFIA and AutoDelfia are not included. We have modified the colours to in the figures.

Point 3: The number of laboratories that use the respective method should be indicated

Response 3: We are sorry for the confusion and we have modified it in the figure legend in red. n=10 or 15 (n equals to the number of distributed DBS per year).  

Point 4: In the results section the author compare and interprete the Performance of different platforms and instruments, however the authors do not provide any calculation or data to prove that the differences they discribe and discuss are really significant.

Response 4: Thank you for your suggestion and we have added the p value in red in the results of the manuscript. We have provided a new figure 2 a-f in which we have shown the difference between groups.

Point 5: The authors should describe, whether and how the EQA program deals with administrative Errors, i.e. mix-up of results during reporting.

Response 5: We appreciate the reviewer’s suggestion. We have provided some details in the discussion of the manuscript. To improve the accuracy of detection in neonatal screening, a platform for quantitative EQA failures was developed for clinical laboratories. The laboratories could automatically analyse the trend of the EQA results through the network reporting system, it could also summarize the unsatisfactory EQA performance for certain years and find the potential reasons, including clerical issues, methodological issues, equipment issues, technical issues, PT process issue, and unexplained issues as well.

We hope that the changes that we have made meet your expectations and we look forward to your response.

Sincerely,

Zhiguo Wang

Natioanl Center for Clinical Laboratories

Beijing Hospital

Beijing, P.R. China

[email protected].

Phone: 86-10-58115054

Fax: 86-10-65273025

Reviewer 2 Report

Quality contol is an important part of all kinds of laboratory work and here the authors describe the system employed in China.

I have the following comments and questions

  1. The Clinet EQA is used for all the evaluations and the adress given as a reference. The system is, however, only described in chinese with chinese symbols.
  2. On line 53 The sending of control samples is descibed but it is unclear if samples were distributed to every lab every year and how many per year.
  3. Line 72-73. Here again it is difficult to understand what is meant. Did a yearly sending of Phe and TSH controls start in 1999 etc?
  4. Figure 2 and Table 1 show the same data and thus Table 1 could be omitted.
  5. The discussion is very short and questions arise. Why did the robust CVs of G6PD and 17-OH - P go up lately. Change of reagents, new unexperienced laboratories joining the program?
  6. The comparison between different analytical methods is probably correct but also contains the influence of time and experience of the lab employing the different methods.
  7. At two places the authors comment that the samples for this project are not equal to blood from newborn babies but the results are only compared with each other in this article.

In conclusion, the authors ought to descrbe the project in more detail and extend the discussion.

Author Response

Thank you for the constructive criticism and the effort spent by the reviewers improving our manuscript. These comments were very helpful. Included please find a point by point reply to the reviewers’ comments. The manuscript has undergone all requested changes with revisions in the revised manuscript highlighted in red.

Point 1: The Clinet EQA is used for all the evaluations and the adress given as a reference. The system is, however, only described in chinese with chinese symbols.

Response 1: Thank you for your question. While the Clinet EQA system are provided only for clinical laboratories in China. That is the reason why the system is only described in Chinese. The good news is that we might provide a bilingual version in the near future.

Point 2: On line 53 The sending of control samples is descibed but it is unclear if samples were distributed to every lab every year and how many per year.

Response 2: We are sorry for the confusion and have clarified the distribution in red in the materials and methods of the manuscript. EQA panels were prepared and each panel contains 5 DBS. Three panels with 15 DBS of Phe and TSH EQA sample were distributed to participating laboratories annually from 2000 to 2019. Two panels with 10 DBS of G6PD EQA sample were distributed annually from 2012 to 2014, and subsequently three panels were distributed annually from 2015 to 2019. Two panels of 17OHP EQA sample were distributed annually from 2013 to 2015, and subsequently three panels were distributed annually from 2016 to 2019. Two panels of amino acid and acylcarnitine by MS/MS EQA program were distributed annually from 2013 to 2019.

Point 3: Line 72-73. Here again it is difficult to understand what is meant. Did a yearly sending of Phe and TSH controls start in 1999 etc?

Response 3: We are sorry for the confusion and have modified it in red in the results of the manuscript. The neonatal screening EQA by testing Phe and TSH was performed in 1999 for the first time. G6PD and 17OHP EQA program were launched since 2012 and 2013 afterwards.

Point 4: Figure 2 and Table 1 show the same data and thus Table 1 could be omitted.

Response 4: We appreciate the reviewer’s suggestion. We have deleted table 1 in the results of the manuscript.

Point 5: The discussion is very short and questions arise. Why did the robust CVs of G6PD and 17-OH - P go up lately. Change of reagents, new unexperienced laboratories joining the program?

Response 5: Thank you for your question and we have explained it in red in the discussions of the manuscript. While the robust CV of G6PD and 17-OHP, on the other hand, went up recently (figure 1), which might be caused by the unexperienced participating laboratories, or the variation of methods and reagents. The finding was reinforced by analysing the data in different measurement system considering the difference of specificity and sensitivity (figure 2).

Point 6: The comparison between different analytical methods is probably correct but also contains the influence of time and experience of the lab employing the different methods.

Response 6:  This is good suggestion. We couldn’t agree more. While it might be hard to determine the time and experience of the lab employing the different methods considering the very different case numbers and time periods. We appreciate the reviewer’s thoughtful suggestion and might perform an investigation involving the influence factors.

Point 7: At two places the authors comment that the samples for this project are not equal to blood from newborn babies but the results are only compared with each other in this article.

Response 7: We appreciate the reviewer’s great suggestion. The of DBS in EQA sample is the limitation of our study.  We tested the homogeneity and stability of our EQA sample, which meet the requirement of CNAS-GL003: Guidance on Evaluating the Homogeneity and Stability of Samples Used for Proficiency Testing. Unfortunately, it’s difficult to compare the EQA sample to clinical sample. We could barely distribute DBS from new-born babies compare the large number of participating laboratories to the shortage in clinical samples.

We hope that the changes that we have made meet your expectations and we look forward to your response.

Sincerely,

Zhiguo Wang

Natioanl Center for Clinical Laboratories

Beijing Hospital

Beijing, P.R. China

[email protected].

Phone: 86-10-58115054

Fax: 86-10-65273025

Reviewer 3 Report

important paper for statistical analysis of methods

abstract language a  bit confusing

what does "11 times  higher" mean?   phe value?  c.v.? please explain that x times higher/greater.

the explanation and figures on the statistical analysis and models are excessive. suggest taking out the formulae on page 2 or otherwise shorten that suggestion as the detail are a bit much for this particular journal.

first two figures on page 3 are hard to read.  perhaps they could be blown up or divided in two or some other way to present more clearly

several english language type corrections needed.  unstable not "instable", for example and first line in discussion "accuracy of screening results could identify...."     doesn't really make sense.  maybe improved accuracy can improve detection or something like that.

Author Response

Thank you for the constructive criticism and the effort spent by the reviewers improving our manuscript. These comments were very helpful. Included please find a point by point reply to the reviewers’ comments. The manuscript has undergone all requested changes with revisions in the revised manuscript in red.

Point 1: abstract language a bit confusing

Response 1: We are sorry for the confusion and have modified it in red in the abstract of the manuscript.

Point 2: what does "11 times higher" mean?   phe value?  c.v.? please explain that x times higher/greater.

Response 2: Thank you for your question. We are sorry for the confusion and have modified it in the results of the manuscript. There was more than a 11-fold increase in the number of the participating laboratories of Phe and TSH EQA program over the last 20 years.

Point 3: the explanation and figures on the statistical analysis and models are excessive. suggest taking out the formulae on page 2 or otherwise shorten that suggestion as the detail are a bit much for this particular journal.

Response 3: We appreciate the reviewer’s suggestion. We have deleted the formulae in the materials and methods of the manuscript.

Point 4: first two figures on page 3 are hard to read.  perhaps they could be blown up or divided in two or some other way to present more clearly.

Response 4: We appreciate the reviewer’s suggestion. We have presented them into table 1.

Point 5: several english language type corrections needed.  unstable not "instable", for example and first line in discussion "accuracy of screening results could identify...."     doesn't really make sense.  maybe improved accuracy can improve detection or something like that.

Response 5: Thank you for your kind revision. We have modified it in the manuscript in red.

We hope that the changes that we have made meet your expectations and we look forward to your response.

Sincerely,

Zhiguo Wang

Natioanl Center for Clinical Laboratories

Beijing Hospital

Beijing, P.R. China

[email protected].

Phone: 86-10-58115054

Fax: 86-10-65273025

Round 2

Reviewer 2 Report

It is interesting to get an insight into the system you have for quality control of the screening methods in China - a vast project containing many variables, which are influenced by many factors.

The manuscript would be clearer if the English would be corrected by an expert.

Author Response

Response to Reviewer Comments

Thank you for the constructive comments and the effort spent by the reviewers improving our manuscript. The suggestions were very thoughtful. Included please find a point by point reply to the reviewers’ comments. The manuscript has undergone English language editing with revisions in the revised manuscript.

Point 1: It is interesting to get an insight into the system you have for quality control of the screening methods in China - a vast project containing many variables, which are influenced by many factors.

Response 1: We appreciate the reviewer’s thoughtful suggestion. In our program, we only collected the analytical methods, reagents or instruments for grouping the data based on the effect of different methods, reagents or instruments. Consider your suggestion, we might perform an survey involving other influence factors, such as the case numbers and time periods etc.

Point 2: The manuscript would be clearer if the English would be corrected by an expert.

Response 2: Thank you for your suggestion. The manuscript has undergone English language editing by MDPI.

We hope that the changes that we have made meet your expectations and we look forward to your response.

Sincerely,

Zhiguo Wang

Natioanl Center for Clinical Laboratories

Beijing Hospital

Beijing, P.R. China

[email protected].

Phone: 86-10-58115054

Fax: 86-10-65273025